# A surge in food insecurity during the COVID-19 pandemic in a cohort in Mexico City

**Luis F. Bautista-Arredondo**[1], **T. Verenice Muñoz-Rocha**[1], **José Luis Figueroa**[2], **Martha M. Téllez-Rojo**[1]\*, **Libni A. Torres-Olascoaga**[1], **Alejandra Cantoral**[3], **Laura Arboleda-Merino**[4], **Cindy Leung**[4,5], **Karen E. Peterson**[4], **Héctor Lamadrid-Figueroa**[6]

1 Center for Nutrition and Health Research, National Institute of Public Health, Cuernavaca, Mexico,
2 Division of Health Economics and Health Systems Innovations, National Institute of Public Health, Cuernavaca, Mexico, 3 Departamento de Salud, Universidad Iberoamericana, Mexico City, Mexico, 4 Nutritional Sciences Department, School of Public Health, University of Michigan, Ann Arbor, Michigan, United States of America, 5 Department of Nutrition, Harvard T.H. Chan School of Public Health, Boston, Massachusetts, United States of America, 6 Department of Perinatal Health, Center for Population Health Research, National Institute of Public Health, Cuernavaca, Mexico

\* mmtellez@insp.mx

## Abstract

### Background

The COVID-19 pandemic has not only caused tremendous loss of life and health but has also greatly disrupted the world economy. The impact of this disruption has been especially harsh in urban settings of developing countries. We estimated the impact of the pandemic on the occurrence of food insecurity in a cohort of women living in Mexico City, and the socioeconomic characteristics associated with food insecurity severity.

### Methods

We analyzed data longitudinally from 685 women in the Mexico City-based ELEMENT cohort. Food insecurity at the household level was gathered using the Latin American and Caribbean Food Security Scale and measured in-person during 2015 to 2019 before the pandemic and by telephone during 2020–2021, in the midst of the pandemic. Fluctuations in the average of food insecurity as a function of calendar time were modeled using kernel-weighted local polynomial regression. Fixed and random-effects ordinal logistic regression models of food insecurity were fitted, with timing of data collection (pre-pandemic vs. during pandemic) as the main predictor.

### Results

Food insecurity (at any level) increased from 41.6% during the pre-pandemic period to 53.8% in the pandemic stage. This increase was higher in the combined severe-moderate food insecurity levels: from 1.6% pre-pandemic to 16.8% during the pandemic. The odds of severe food insecurity were 3.4 times higher during the pandemic relative to pre-pandemic levels (p<0.01). Socioeconomic status quintile (Q) was significantly related to food insecurity (Q2 OR = 0.35 p<0.1, Q3 OR = 0.48 p = 0.014, Q4 OR = 0.24 p<0.01, and Q5 OR = 0.17

**Data Availability Statement:** The data underlying the results presented in the study are available from Deep Blue Data Repository at https://doi.org/10.7302/jm7j-9v49.

**Funding:** KEP and MMTR. This work was supported by the National Institutes of Environmental Health Sciences at the National Institutes of Health (P01ES022844, R24ES028502), the US Environmental Protection Agency (RD83543601). The funders did not play any role in the conception or writing of this paper (or any others from the ELEMENT Project). The funders had no role in study design, data collection and analysis, decision to publish, or preparation of the manuscript.

**Competing interests:** The authors have declared that no competing interests exist.

p<0.01), as well as lack of access to social security (OR = 1.69, p = 0.01), and schooling (OR = 0.37, p<0.01).

## Conclusions

Food insecurity increased in Mexico City households in the ELEMENT cohort as a result of the COVID-19 pandemic. These results contribute to the body of evidence suggesting that governments should implement well-designed, focalized programs in the context of economic crisis such as the one caused by COVID-19 to prevent families from the expected adverse health and well-being consequences associated to food insecurity, especially for the most vulnerable.

## Introduction

The COVID-19 pandemic has unfolded in turn into a major international economic crisis [1]. Mobility restrictions implemented in most countries to prevent the spread of the SARS-CoV2 virus affected labor and income. Globally, more than 300 million full-time jobs were lost by the second quarter of 2020, compared to 130 million in the same period of 2019 [2]. In Mexico, one of the countries more affected by the pandemic in terms of the number of case-fatality ratio and deaths per 100,000 inhabitants [3], mobility restrictions started in March 2020 with the governmental "stay at home" campaign and the closing of in-person schooling and non-essential economic activities [4]. In 2020, the country experienced an 8.5% decrease in GDP [5], and direct consequences for households' livelihoods: 21% of Mexican households had at least one family member lose their job, and had reductions in labor income [6, 7]. Estimates from 2021 suggest that unemployment and labor recovered comparing to 2020, but despite this recovery, unemployment and labor have not reached pre-pandemic levels, affecting especially families in the informal sector of the economy [8]. The loss of employment and economic wellbeing as a result of the pandemic is likely to affect a household's food security in Mexico as suggested by previous research that documents a rise in the number of food insecure households, particularly affecting families that were not vulnerable before the pandemic [9, 10].

Food security is achieved when "all people, at all times, have physical and economic access to sufficient, safe and nutritious food that meets their dietary needs and preferences for an active and healthy life" [11]. This definition comprises four dimensions: food availability, food access, utilization, and stability. Alterations in any of the food security dimensions can result in a reduction of the quality and quantity of food consumed by household individuals; such limitations affect diet and nutritional status, and therefore health and development [12, 13]. In the context of the pandemic, the rise of food prices, food system disruptions and loss of income has translated into difficulties in access to healthy foods, which in turn might exacerbate the consequences on family food security [14].

Among the many economic and social consequences that came with this crisis, food insecurity has been showing a stark increase worldwide [15]. More than 135 million people have been at serious risk of being chronically food insecure due to the COVID-19 pandemic, and predictions from the World Food Program suggest that this number could rise as the crisis continues to affect the economic conditions of families. In Mexico, the percentage of any level of food insecurity increased between 2018 and 2020 from 55.5% to 59.1% (data collected between August and November 2020) [7].

However, in Mexico, there is scarce pre-COVID longitudinal data on the effect of the pandemic on food insecurity. This study aimed to estimate the impact of the COVID-19 pandemic on the occurrence of food insecurity in an existing cohort of women living in Mexico City, and the socioeconomic characteristics associated with food insecurity severity. In this work, we tested the hypothesis that the COVID-19 pandemic caused a sudden increase in food insecurity in the sample from 2015–19 to 2020–21.

## Methods

### Study population and sample

We analyzed data from a panel of 685 Mexico City women in the ELEMENT cohort at three time points between 2015 and 2021. Details of the ELEMENT cohort have been described elsewhere [16]. In summary, the Early Life Exposure in Mexico to ENvironmental Toxicants (ELEMENT) Project is a mother–child pregnancy and birth cohort initiated in the mid-1990s to explore the effects of exposure to lead and other environmental toxicants on infant´s health and development. The Project is based on three sequentially recruited distinct cohorts of mother-child pairs recruited from a homogenous low-income to middle-income population attending family clinics in Mexico City. These cohorts comprise 1643 mother-child pairs, some of whom we have followed for almost three decades. The specific objectives of each cohort are different, but the background and sociodemographic characteristics of the three original cohorts are similar. Following these cohorts, several follow-up studies have been conducted for participating mothers and children of the original cohorts. The same research and data collection have followed all cohorts, and the same facilities and protocols were used.

Data for this analyses was collected as part of one of ELEMENT´s follow-up studies during in-person visits to the American British Cowdray (ABC) Hospital research center on two pre-COVID time points in the 2015–2019 period, and by telephone call during the COVID-19 pandemic in 2020. The first visit (T1) occurred between March 2015 –August 2017, the second visit (T2) took place between November 2016-January 2019, and the third visit (COV) was by telephone between June 2020 –March 2021, during the COVID-19 pandemic. The number of interviewed women was 552 for T1, 519 for T2, and 590 for COV; 432 women were interviewed on all three time points, and 685 women were interviewed at one time point at least.

### Assessment of outcome and covariates

Information on food insecurity was assessed using the Latin-American Scale of Food Security (ELCSA) [17], a measure focused in household report of food access and utilization. The scale includes 15 questions, of which eight refer directly to food insecurity either in the household or specifically in the adults within the household; and seven referred to food insecurity in children (<18 years old). The reference time frame was three months prior to survey administration. Based on the number of positive responses, and depending on whether or not households included children, households were grouped into the following categories: In households without children: no food insecurity (0 positive responses), mild (1–3), moderate (4–6), and severe food insecurity (7–8). In households with children: no food insecurity (0 positive responses), mild (1–5), moderate (6–10), and severe food insecurity (11–15) [17].

Socioeconomic status (SES) of the family of the respondent was estimated using a wealth index (or asset index) based on measures of housing (constructing materials used for flooring), number of rooms, type of toilet, electricity, and ownership of assets (water-heater, TV, computer, radio, stove, washing machine, microwave, and vacuum cleaner). To generate the index, we used principal component analysis as described elsewhere [18–20]. Scores from the first component, which explained 39.2% of the total variance, were used to create the wealth

variable. We included those assets and services recommended in the literature and that were available in our questionnaire [19, 21]. For analysis, the wealth index was categorized into SES quintiles. As stated in the literature, this index provides a measure of relative wealth of the families in our sample. That is, it allows us to compare only how vulnerable these families are in relation to other families in the cohort only.

Other covariates on which information was collected by questionnaire included age of the respondent; maximum schooling of a household member (Middle School or less, High School, Professional or more); access to social security (Yes/No); marital status (married or cohabitating vs. single/divorced/widowed); identification of the head of the household (woman respondent, husband or partner, shared between respondent and husband, son, brother, grandparents, brother-in-law, other); whether the woman respondent had a job over the previous week, (Yes, No, No, I do housework); food consumption perception in the last month (Equal, Higher, Less); and whether the household included children under 18 years of age (Yes/No).

### Statistical analysis

Descriptive statistics of the study variables were analyzed, over the whole period and, in order to estimate changes between pre-pandemic and pandemic periods, by time of data collection. Additionally, a bivariate comparison of food insecurity and covariates by SES quintiles was performed.

To evaluate periodic fluctuations in food insecurity during the study period, we modeled the expected value of the food insecurity scale (with values 0 to 3, where 0 represents "no food insecurity" and 3 represents "severe food insecurity") both as a function of calendar time and by data collection. We also modelled the expected value of the food insecurity scale as a function of the day of the year, to evaluate possible seasonal patterns of food insecurity, using means of kernel-weighted local polynomial regressions.

In order to estimate the impact of the COVID-19 pandemic on food insecurity, we fitted random and fixed-effects ordered logistic regression models of food insecurity to obtain odds ratios and 95% confidence intervals. Timing of data collection (pre-pandemic [T1 and T2] vs. during pandemic [COV]) was the main predictor; the effect of the pandemic was defined as the Odds Ratio (OR) for COV (2020) vs. the T1 time point (2015–2017). Potential covariates were included in models according to background knowledge of food insecurity drivers. Final models included month of the year, SES quintiles, access to social security, and schooling as covariates for the random-effects model, and included months of the year for the fixed effects model. Current employment was not included in any model as it was deemed an intermediate variable. Models eventually excluded the marital status covariate as it presented low variation (less than 10% of the households had a single mother) and was not significantly related to the outcome. We present both fixed- and random-effects ordinal logistic regression models as a robustness check. Although fixed-effects models are theoretically known to be consistent, and thus likely preferable, the advantage of also presenting random-effects models is that they allow for time-fixed regressors to be included in the model. All statistical analyses were performed using Stata v. 16.0 [22].

We obtained written informed consent from study subjects and data collection procedures were reviewed and approved by the Research Ethics Committee of the National Institute of Public Health, Mexico.

## Results

During the pandemic, the average age of women in the cohort was approximately 47 years. Nearly 50% of the respondents considered themselves to be the head of the household, either

**Table 1. Characteristics of the sample during the pandemic in COV visit, by socioeconomic index quintile.**

| Characteristics | | | | | | | |
|---|---|---|---|---|---|---|---|
| Mother´s age (years) µ(SD) | **46.6(5.6)** | | | | | | |
| | % | \multicolumn — Quintiles of the Socio Economic Index | | | | | |
| | | 1 | 2 | 3 | 4 | 5 | p-value |
| **Maximum schooling at home** | | | | | | | |
| Middle School or less | 21.1 | 32.6 | 18.2 | 13.6 | 11.2 | 6.5 | |
| High School | 54.8 | 58.4 | 59.1 | 69.3 | 55.1 | 48.9 | <0.001 |
| Professional or more | 24.1 | 9.0 | 22.7 | 17.1 | 33.7 | 44.6 | |
| **Access to social security** | | | | | | | |
| Yes | 64.4 | 52.8 | 60.9 | 74.7 | 57.3 | 76.9 | 0.001 |
| **Marital status** | | | | | | | |
| Married | 72.7 | 61.5 | 64.1 | 74.7 | 75.9 | 82.5 | |
| Cohabitating | 17.4 | 24.4 | 23.1 | 16.0 | 10.8 | 12.5 | |
| Single/divorced | 9.9 | 14.1 | 12.8 | 9.3 | 13.3 | 5.0 | 0.072 |
| **Head of the household** | | | | | | | |
| Mother interviewed | 24.1 | 28.1 | 27.3 | 29.6 | 22.5 | 21.7 | |
| Husband or Partner | 35.6 | 37.1 | 36.4 | 35.2 | 30.3 | 31.5 | 0.338 |
| Mother and husband | 25.9 | 16.8 | 28.4 | 22.7 | 34.8 | 29.4 | |
| Son, brother, grandparents, brother-in-law, other | 14.4 | 18.0 | 7.9 | 12.5 | 12.4 | 17.4 | |
| **Had a job in the last week** | | | | | | | |
| Yes | 57.1 | 53.9 | 57.9 | 55.2 | 62.9 | 55.4 | |
| No | 16.8 | 12.4 | 17.1 | 21.8 | 13.5 | 19.6 | 0.263 |
| No, I do housework | 26.1 | 33.7 | 25 | 23.0 | 23.6 | 25.0 | |
| **Households with children <18** | | | | | | | |
| Yes | 66.8 | 70.8 | 75.0 | 64.8 | 62.9 | 60.9 | 0.237 |

by themselves or co-heading with their spouse-partner; over 90% reported cohabitation with a spouse-partner (72.7% married and 17.4% cohabitating). Close to 60% had a job, nearly two-thirds of the participants had at least high-school education, and close to one fifth had at least a bachelor's degree. Complete descriptive statistics of the study population at COV visit are shown on Table 1.

With respect to the food insecurity levels, from T1 (2015–2017) to T2 (2016–2019), the proportion of households with any level of food insecurity decreased from 42.2% to 40%. Similarly, the proportion of households with moderate food insecurity decreased from 3.4% to 1%, and the proportion of households with severe food insecurity remained the same from 0.5% to 0.6%. Food insecurity during the COV time point was significantly higher compared to both earlier time points. Most households (53.8%) reported some level of food insecurity, while 9.0% and 7.8% reported moderate and severe levels, respectively (Table 2).

During the pandemic, almost 70% of households in the lowest SES quintile experienced any level of food insecurity: severe food insecurity was as high as 15.7%, while moderate and mild food insecurity accounted for 11.2% and 42.7%, respectively. (Table 3) The proportion of participants reporting severe insecurity was significantly higher in the less educated and in those without access to social security (Table 3).

The expected value of the food insecurity scale was almost constant throughout T1 and T2 with an average close to 0.5. However, the trajectory of the insecurity scale during the COV data collection round had a distinct inverted u-shaped form. Although levels around June 2020 and January 2021 were similar to values seen at T1 and T2, the average value peaked at

**Table 2. Proportion of food insecurity by level and study visit.**

| Variables | T1 2015–2017 | T2 2016–2019 | COV 2020–2021 |
|---|---|---|---|
| | N = 522 | N = 519 | N = 590 |
| | % | % | % |
| **No insecurity** | **53.8** | **58.4** | **46.2** |
| **Food insecurity (any level)** | **46.2** | **41.6** | **53.8** |
| Mild | 42.2 | 40.0 | 37.0 |
| Moderate | 3.5 | 1.0 | 9.0 |
| Severe | 0.5 | 0.6 | 7.8 |

Result of the **Fisher's exact test**, T1 vs COV p-value <0.001, and T2 vs COV p-value <0.001

close to 1 around November 2020, almost double the value at the same time of year during the T1 and T2 collections (Fig 1).

Results of the ordered logit models of food insecurity are shown in Table 4. The odds of having greater food insecurity during the COV collection period was over 3-fold relative to T1 (OR 3.43, p<0.01) regardless of the choice of model (fixed-effects vs. random-effects), and the odds of having greater food insecurity during COV visit was 4.5 for fixed-effects and 3.7 for random-effect relative to T2 (p<0.01) (values not showed in the tables). Protective factors for food insecurity included higher SES quintiles (OR = 0.36 for SES quintile 2, OR = 0.48 for SES quintile 3, OR = 0.24 for SES quintile 4, and OR = 0.17, for SES quintile 5, all p<0.01), and schooling at the professional level (OR = 0.37, p<0.01), whereas lack of access to social security was associated with higher food insecurity (OR = 1.69, P = 0.01).

**Table 3. Bivariate comparison of food insecurity and covariates by quintiles of the socio economic index during the COV visit (n = 443).**

| Variables | No food insecurity | Mild | Moderate | Severe | p-value |
|---|---|---|---|---|---|
| | % | % | % | % | |
| **SES (quintiles)** | | | | | |
| 1 | 30.4 | 42.7 | 11.2 | 15.7 | |
| 2 | 40.2 | 42.5 | 6.9 | 10.4 | |
| 3 | 35.6 | 43.7 | 14.9 | 5.8 | 0.001 |
| 4 | 52.8 | 30.4 | 6.7 | 10.1 | |
| 5 | 61.5 | 27.5 | 8.8 | 2.2 | |
| **Maximum schooling at home** | | | | | |
| Middle School or less | 31.0 | 46.5 | 8.4 | 14.1 | |
| High School | 40.5 | 37.1 | 12.0 | 10.4 | <0.001 |
| Professional or more | 61.0 | 31.9 | 5.3 | 1.8 | |
| **Access to social security** | | | | | |
| Yes | 47.5 | 37.8 | 7.7 | 7.0 | |
| No | 38.2 | 36.3 | 13.4 | 12.1 | 0.038 |
| **Marital status** | | | | | |
| Married | 45.6 | 37.9 | 8.5 | 8.0 | |
| Cohabitating | 38.1 | 41.7 | 10.7 | 9.5 | |
| Single/divorced | 52.1 | 35.4 | 6.3 | 6.2 | 0.812 |

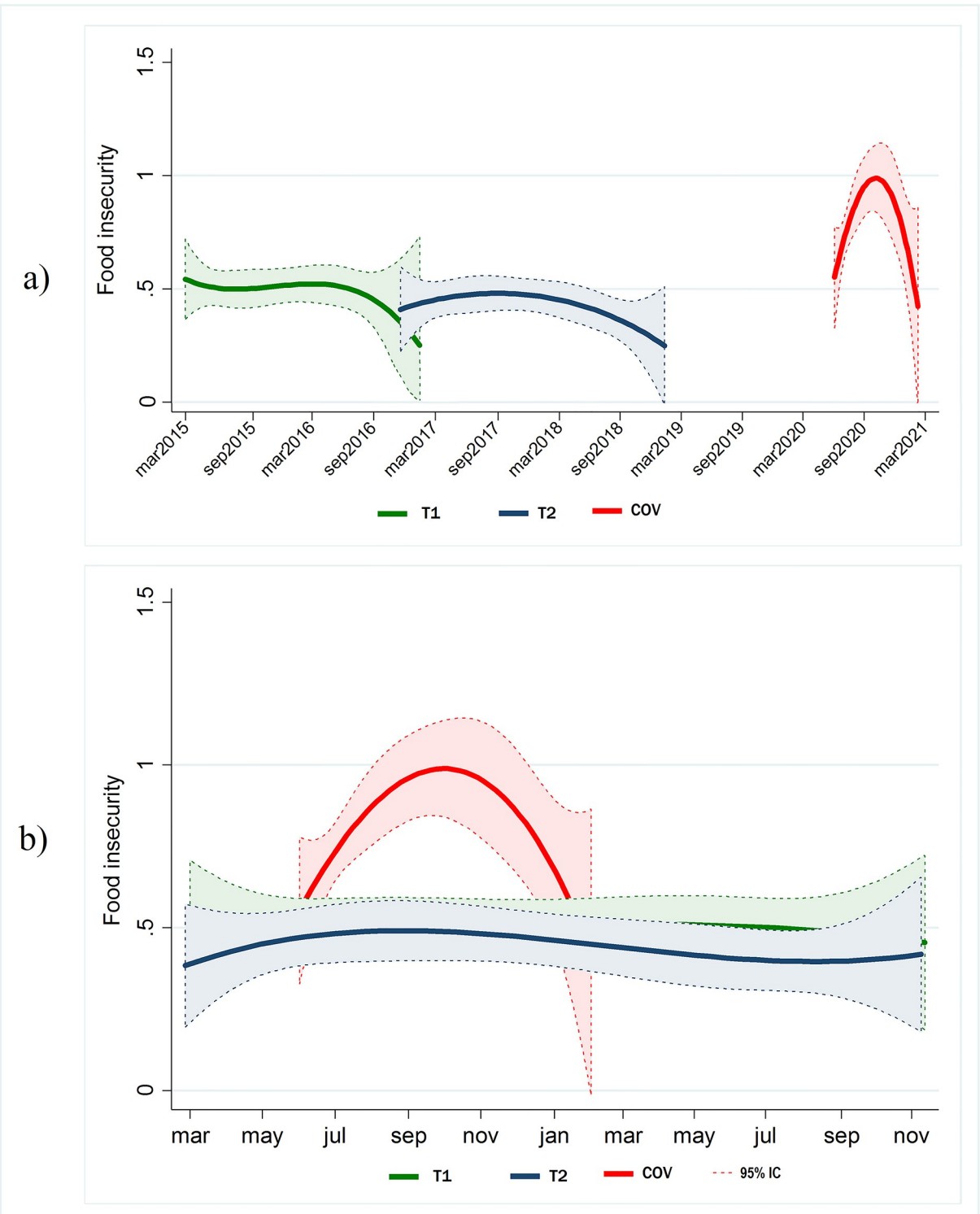

**Fig 1. Fluctuations in the expected value of the food insecurity scale as a function of time.** By data collection wave, with 95% confidence interval: a) As a function of calendar time; b) As a function of the day of the year.

**Table 4. Random and fixed-effects ordered logistic regression models of food insecurity to obtain odds ratios and 95% confidence intervals.**

| Variables | MODEL 1* | | | MODEL 2** | | |
|---|---|---|---|---|---|---|
| | Odds Ratio | (95% CI) | p- value | Odds Ratio | (95% CI) | p-value |
| **Visit** | | | | | | |
| T1 | Ref | | | Ref | | |
| T2 | 0.760 | (0.582, 0.992) | 0.044 | 0.788 | (0.580, 1.073) | 0.130 |
| COV | **3.433** | (2.348, 5.019) | <0.001 | **3.025** | (2.108, 4.342) | <0.001 |
| **SES (quintiles)** | | | | | | |
| 1 | | | | Ref | | |
| 2 | | | | **0.357** | (0.198, 0.644) | 0.001 |
| 3 | | | | **0.481** | (0.267, 0.864) | 0.014 |
| 4 | | | | **0.243** | (0.132, 0.445) | <0.001 |
| 5 | | | | **0.171** | (0.091, 0.324) | <0.001 |
| **Maximum schooling at home** | | | | | | |
| Middle School or less | | | | Ref | | |
| High School | | | | 0.902 | (0.535, 1.520) | 0.698 |
| Professional or more | | | | **0.374** | (0.198, 0.705) | 0.002 |
| **Access to social security** | | | | | | |
| Yes | | | | **Ref** | | |
| No | | | | **1.687** | 1.133, 2.511) | 0.010 |
| **Months** | | | | | | |
| January | Ref | | | **Ref** | | |
| February | 0.621 | (0.226, 1.705) | 0.355 | 1.185 | (0.487, 2.883) | 0.708 |
| March | 0.596 | (0.250, 1.420) | 0.243 | 0.836 | (0.367, 1.897) | 0.668 |
| April | 1.901 | (0.851, 4.247) | 0.117 | 1.793 | (0.803, 4.003) | 0.154 |
| May | 2.127 | (0.865, 5.226) | 0.100 | 2.276 | (1.007, 5.141) | 0.048 |
| June | 1.021 | (0.461, 2.257) | 0.959 | 1.075 | (0.521, 2.218) | 0.845 |
| July | 1.598 | (0.710, 3.594) | 0.257 | 1.473 | (0.716, 3.027) | 0.292 |
| August | 0.765 | (0.314, 1.859) | 0.554 | 0.730 | (0.331, 1.608) | 0.435 |
| September | 0.878 | (0.341, 2.254) | 0.787 | 1.021 | (0.450, 2.316) | 0.961 |
| October | 1.612 | (0.728, 3.570) | 0.239 | 1.859 | (0.860, 4.021) | 0.115 |
| November | 1.302 | (0.541, 3.130) | 0.555 | 1.949 | (0.851, 4.466) | 0.114 |
| December | 0.825 | (0.307, 2.211) | 0.702 | 1.152 | (0.460, 2.917) | 0.765 |

* **Fixed Effects** Ordinal Regression model, adjusted by month of the year in which the mother came to visit (stage T1 and T2), and when she received the COV visit (n = 685).

**Random effects** Ordinal Regression model, adjusted for quintiles of SES, maximum schooling at home, access to social security, and month of year (n = 443).

The response variable is categorized as follows: 0-no insecurity, 1-mild, 2-moderate and 3-severe.

## Discussion

COVID-19 pandemic increased food insecurity among participants of the ELEMENT cohort study, living in Mexico City, one of the most affected urban areas in the world by the pandemic in terms of COVID-19 fatalities [23]. We find larger effects for households at the lower end of the wealth distribution.

Before the COVID-19 pandemic, almost all of the study population was distributed between food secured (58.4%) and mild food insecurity (40.1%); and a very low proportion of households had moderate and severe food insecurity, as the cumulative proportion for these two levels was 1.5%. In contrast, the National Health and Nutrition Survey (ENSANUT 2018–19), a

representative cross-sectional national survey, reported 19% and 22.7% of combined moderate-severe food insecurity in Mexico City and at the national level respectively at pre-pandemic times [24]. The ENSANUT 2020 collected data during the COVID-19 pandemic (August-November 2020 period) and, compared to 2018, reported a downward trend in the combined moderate-severe food insecurity, with a decrease of 4.6 and 2.1 percentage points in Mexico City and at the national level respectively (Mexico City: 14.4%; Mexico: 20.6%) [7], which suggests that the pandemic did not adversely affected the more serious levels of this indicator. In contrast, our results suggest an important impact of the pandemic on moderate-severe food insecurity in our study population, reaching 16.8% in 2020; a more than ten-fold increase with respect to the pre-COVID period.

Additional information on food insecurity in Mexico is available in the ENCOVID-19 survey (COVID-19 effects on the Well-being of Mexican Households Survey), a nationally representative cross-sectional survey conducted monthly by telephone [9]. This survey shows a national average of 31.4% of combined moderate-severe food insecurity for the period May 2020-March 2021; ENCOVID-19 did not report data from Mexico City. This figure is almost double the prevalence of combined moderate-severe food insecurity documented in our study population, and nearly 50% higher than the prevalence reported in ENSANUT 2020. It is important to highlight that both ENSANUT 2020 and ENCOVID-19 results are based on cross-sectional information, which only permits to assess if the overall prevalence of insecurity increased across rounds of information. Our results are based on panel information, which allowed us to control for bias stemming from non-observable characteristics.

Our results are not meant to present nationally or region-specific representative statements about food insecurity, but to suggest that food insecurity increased within a closed cohort with panel data and that certain demographic characteristics were identified in making a household more vulnerable to increased food insecurity during COVID. In this regard, there are a number of factors we should state as a potential source of difference between our results and published survey data in Mexico City and at the national level respectively before the pandemic [24]. The ELEMENT study population is, on average, wealthier than a representative sample of the country as a whole, but also in comparison to Mexico City. The pre-pandemic SES status of the original cohort of the ELEMENT study was assessed using the methodology of the *Asociación Mexicana de Agencias de Inteligencia de Mercado y Opinion (AMAI)* [25]. Given the scope of our analysis, we used a new SES index with information on characteristics at the time of the COVID collection phase, based on Principal Component Analysis. This new measure, though, is a relative measure of wealth, providing information on wealth differences across households in the study sample only. However, it is possible to infer from data based on the AMAI index how these households compare in the context of the population in Mexico City. Our study population is composed of a non-representative sample of families from mid to low socioeconomic level, but almost nonexistent representation of the lowest socioeconomic level households in Mexico City. Our results suggest that these urban families were especially affected by the crisis created by the pandemic, a phenomenon that has not been, to our knowledge, captured in other studies in Mexico before. According to national data, labor was severely affected by the pandemic lockdown in this population groups [26].

Results show that after a stark increase that peaked in November 2020, average food insecurity decreased to pre-pandemic levels. Our results, however do not capture effects after November 2020, just when a newly third peak of the epidemic in Mexico started to evolve and lockdown measures were not as strict as before. Thus, it is expected that a new increase in food insecurity could affect these families as shown by ENCOVID19, which collected data up to March 2021.

Our results show evidence of a significant food insecurity pandemic effect in our study sample, and the described return to prior levels of food insecurity in November 2020 should be interpreted with caution, as according to literature, even transient increases in the prevalence of mild and moderate food insecurity can lead to adverse health outcomes. For example, in order to ensure caloric intake of all household members during times of resource scarcity, individuals may choose to buy and consume low-cost energy-dense foods, many of which are high in added sugars, fats and sodium and low in nutritional content. In parallel, there is a reduction in the variety of foods and nutrients consumed, particularly fruits, vegetables and some micronutrients [27, 28]. Together, these behaviors compromise diet quality and promotes visceral fat accumulation and weight gain [12, 28]. Research evidence points to a significant association between food insecurity and increased risk of developing diet-related chronic diseases, including diabetes, hypertension, and cardiovascular disease [28]. All age groups can be affected by food insecurity: in children it can impact health and development, including behavior outcomes and school performance [13]; adults are prone to develop diet-sensitive chronic diseases [28], and in older adults affects physical and mental health, and quality of life [12].

We did not find differences in member household composition as reported by national estimates. Both ENSANUT and ENCOVID report higher food insecurity levels in households with minors. However, ENCOVID used a modified version of the ELCSA that does not consider the household composition by design. We didn't conduct this analysis, because we used the original version of the instrument that accounts for household composition by design.

The food security definition and its related dimensions: food availability, food access, utilization and stability are notions widely accepted by policymakers and international stakeholders. Countries need to promote practical policy efforts to tackle risks across these four dimensions to ensure food security in their population. However, policy responses across countries to counterbalance the effects of the pandemic have been heterogeneous. For example, many developed economies adopted policies to incentivize the economy and protect families from sliding into food insecure conditions. Other countries, like China, prioritized supply-side responses to ensure food production and availability in the form of subsidies or different types of fiscal incentives [29]. In Latin America, most countries put in place policy efforts to facilitate economic and physical access to food. A notable exception was observed in Mexico, where governmental actions merely reinforced food production programs in rural areas and early cash transfers to families enrolled in pre-pandemic social programs [30]. Insufficient policy responses and a stark rise in food prices in Mexico in 2021. Mexico closed in 2021 with a yearly inflation rate of 7.36%, the highest in 20 years. According to the National Statistics Institute of Mexico, the rise of food prices was the main contributor to the observed inflation exacerbate the country's risk of food insecurity in 2022 [31].

The main strength of our study is the longitudinal panel data collected at two time points before the pandemic, and one-time point during the COVID pandemic. Within the study, data were also available to examine the effect on food insecurity for households with different economic and social characteristics, like the socioeconomic status of the family, employment status, and access to healthcare public services. Our study has also some limitations that has to be addressed. Our study population is not national or state representative, which prevent us from generalizing our results to the whole country or city. However, given the longitudinal study design, our results are able to make important inferences on the effect of the pandemic on food insecurity within this population.

Our study provides robust evidence on a significant impact of COVID-19 in food security status in a mid to low socioeconomic sample in Mexico City, that affected greatly the more disadvantaged quintiles of the SES index. Our findings contribute to the growing body of

evidence suggesting that well-designed and focalized programs are especially crucial during pandemics or natural disasters, to prevent families from the adverse health and well-being effects associated with food insecurity.

## Acknowledgments

We thank the American British Cowdray Hospital in Mexico City for providing the facilities used to conduct this research.

## Author Contributions

**Conceptualization:** Luis F. Bautista-Arredondo, José Luis Figueroa, Martha M. Téllez-Rojo, Héctor Lamadrid-Figueroa.

**Data curation:** T. Verenice Muñoz-Rocha, Héctor Lamadrid-Figueroa.

**Formal analysis:** T. Verenice Muñoz-Rocha, Héctor Lamadrid-Figueroa.

**Funding acquisition:** Martha M. Téllez-Rojo, Karen E. Peterson.

**Investigation:** Luis F. Bautista-Arredondo, José Luis Figueroa, Martha M. Téllez-Rojo, Libni A. Torres-Olascoaga, Laura Arboleda-Merino, Cindy Leung, Karen E. Peterson, Héctor Lamadrid-Figueroa.

**Methodology:** Luis F. Bautista-Arredondo, T. Verenice Muñoz-Rocha, José Luis Figueroa, Martha M. Téllez-Rojo, Libni A. Torres-Olascoaga, Cindy Leung, Karen E. Peterson, Héctor Lamadrid-Figueroa.

**Project administration:** Luis F. Bautista-Arredondo, Martha M. Téllez-Rojo, Libni A. Torres-Olascoaga, Laura Arboleda-Merino, Karen E. Peterson.

**Resources:** Martha M. Téllez-Rojo, Libni A. Torres-Olascoaga, Laura Arboleda-Merino, Karen E. Peterson.

**Software:** T. Verenice Muñoz-Rocha, Héctor Lamadrid-Figueroa.

**Supervision:** Luis F. Bautista-Arredondo, Martha M. Téllez-Rojo, Libni A. Torres-Olascoaga, Laura Arboleda-Merino, Karen E. Peterson, Héctor Lamadrid-Figueroa.

**Validation:** Luis F. Bautista-Arredondo, José Luis Figueroa, Martha M. Téllez-Rojo, Libni A. Torres-Olascoaga, Alejandra Cantoral, Laura Arboleda-Merino, Cindy Leung, Karen E. Peterson, Héctor Lamadrid-Figueroa.

**Visualization:** T. Verenice Muñoz-Rocha, Héctor Lamadrid-Figueroa.

**Writing – original draft:** Luis F. Bautista-Arredondo, T. Verenice Muñoz-Rocha, José Luis Figueroa, Martha M. Téllez-Rojo, Libni A. Torres-Olascoaga, Laura Arboleda-Merino, Héctor Lamadrid-Figueroa.

**Writing – review & editing:** Luis F. Bautista-Arredondo, T. Verenice Muñoz-Rocha, José Luis Figueroa, Martha M. Téllez-Rojo, Libni A. Torres-Olascoaga, Alejandra Cantoral, Laura Arboleda-Merino, Cindy Leung, Karen E. Peterson, Héctor Lamadrid-Figueroa.

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
