## [Decision Letter · Decision Letter 0]

12 Jul 2022

PONE-D-22-12889A surge in food insecurity during the COVID-19 pandemic in a cohort in Mexico CityPLOS ONE

Dear Dr. Tellez-Rojo,

Thank you for submitting your manuscript to PLOS ONE. After careful consideration, we feel that it has merit but does not fully meet PLOS ONE’s publication criteria as it currently stands. Therefore, we invite you to submit a revised version of the manuscript that addresses the points raised during the review process.

I want to thank the authors for making this submission, and their contribution to the literature on this issue of sustained significance in the context of a continuing COVID-19 pandemic. The reviews returned a significant spread with regards to the suitability/generalizability of the findings presented herein. I urge careful consideration of those comments especially ones relating to methodological limitations, fit to existing literature, and impact of results in your revision. 

We look forward to receiving your revised manuscript.

Kind regards,

Courtney Elizabeth Heffernan

Academic Editor

PLOS ONE

Journal Requirements:

“KEP and MMTR. This work was supported by the National Institutes of Environmental Health Sciences at the National Institutes of Health (P01ES022844, R24ES028502), the US Environmental Protection Agency (RD83543601). The funders did not play any role in the conception or writing of this paper (or any others from the ELEMENT Project).”

Reviewers' comments:

Reviewer's Responses to Questions

**Comments to the Author**

1. Is the manuscript technically sound, and do the data support the conclusions?

Reviewer #1: Yes

Reviewer #2: Partly

Reviewer #3: Partly

2. Has the statistical analysis been performed appropriately and rigorously? 

Reviewer #1: Yes

Reviewer #2: No

Reviewer #3: Yes

3. Have the authors made all data underlying the findings in their manuscript fully available?

Reviewer #1: Yes

Reviewer #2: No

Reviewer #3: Yes

4. Is the manuscript presented in an intelligible fashion and written in standard English?

Reviewer #1: Yes

Reviewer #2: Yes

Reviewer #3: Yes

5. Review Comments to the Author

Reviewer #1: I have published a number of studies regarding the measurement of food insecurity, and I feel like the authors did an adequate, even good, job. There was a clear objective, the data allow them to achieve their objective, and they used scientifically valid statistical methods. I appreciate the authors keeping the manuscript succinct. For example, they could have spent pages and pages discussing the ELEMENT cohort but they prudently remark instead that it is described elsewhere and they provide that citation.

A couple of remarks

Use of the term “surge” in the title doesn’t bother me but most reviewers do not like that kind of language.

On line 126, what is a “secular” fluctuation in food security? May be a translation issue here.

On lines 136 – 137 you remark that you only retain covariates that show statistical significance. In my field that is highly discouraged and is referred to as a form of “p hacking”. People still do it all the time, you are just one of the few authors who admit it! If I were you though I would instead put your original estimations that also include the insignificant coefficients, lest someone use your paper as an example of what not to do. This isn’t a huge deal for me but it is for some in my field.

On Table 1 I don’t understand what the COV% column means.

Reviewer #2: Review for Article PONE-D-22-12889, Titled, A surge in food insecurity during the COVID-19 pandemic in a cohort in Mexico City

Summary

This study’s goal is to estimate the impact of the COVID-19 pandemic on the occurrence of food insecurity in a cohort of women living in Mexico City, and the socioeconomic characteristics associated with food insecurity severity. I can definitely see this paper making contribution to still ongoing issue that countries face due to COVID so it is a very relevant topic. However, the manuscript could be stronger with a few suggestions listed below:

Abstract

Typo in the abstract, Results portion

Study population and sample

The timings of the visits are difference across the years. Is that a concern? How can that impact results depending on what season of the year/crop cycle it is in?

This should be written differently. “A total of 685 women had data on food security at least one data time point, while 432 women had information on all three time points.”

Results

Whereas all other types of food insecurity seem to have increased, it is puzzling that Mild food

insecurity decreased. This is also mentioned the in abstract. More explanation and intuition here could be helpful.

Line 131: missing comma

Table 1: since determinant seems to be pre and post pandemic timing, it would be better to present table 1 by the survey rounds, rather than quintiles (similar to table 2)

Line 177: missing “the”

Figure 1 does not appear where its placed.

Econometric method

Usually there are strong reasons behind why authors would choose to use FE versus RE models. In this paper, there is no justification or mention of why both are done? Also, there are no robustness checks of the models.

Since the month of data collection is different, I thought I saw somewhere that authors control for that, however, I don’t see that in Table 4. And how does controlling for months correlate with the survey rounds themselves? What is the correlation?

The authors state that they used a host of characteristics but they only present the ones that are statistically significant. Are the rest of the variables not presented because authors are running out of space in the paper? Mention of these variables in table notes would at least be expected so readers can see it at a glance.

Discussion

“To our knowledge, this is the first study with panel data that estimated the magnitude of the COVID-19 impact on food insecurity in Mexico”. I might be wrong but I thought that the World Bank did it for many countries across the world and that should have included Mexico.

Line 217: grammar

Line 267: grammar

Line 269: the definition should be written early on since the study is about food insecurity

Reviewer #3: The manuscript addresses a relevant issue for which evidence already exists in Mexico and other countries. Still, it contributes from a methodological perspective by using data from a panel study and incorporating a longitudinal analysis.

Nevertheless, the article presents several problems regarding the representativeness of its results, context, dialogue with specialized international literature, hypothesis, and the SES variable.

1) Representativeness of results. The authors work with a cohort of women recruited from maternity hospitals in Mexico City (following the cited paper because it is not mentioned in this manuscript). How this cohort represents information about the general population? Why study this group could be interesting for understanding food insecurity? What are the limits of this study concerning this sample? The authors should explain more details about the recruitment, deep sample characteristics, and their limitations.

The limits of this research imply that the authors must moderate some statements. For example, the abstract mentions that food insecurity soared in Mexico City in 2020 (in general), but in the same abstract, they mention that they work with data from mid and lower mid to low SES in Mexico City. If their sample came from this group of women, they must be cautious with generalizations.

2) Context. The authors should give information about the Mexican contexts to understand the results better. What is Mexico's long-term situation of food insecurity (and obesity)? Does Mexico is in a nutritional transition? How many people have formal and informal employment? How is this situation different or like other Latin American countries?

3) Dialogue with specialized literature. The reference to the literature in the introduction is weak. Later in the discussion section, the results are primarily confronted with the Mexican surveys. That implies several problems. First, it does not allow situated the food insecurity during COVID as a world phenomenon. Second, it does not give justification to justify the variables used in the models.

For example, in the introduction, the authors mentioned in a very general way that various variables can affect food insecurity without specifying the variables used in their work. In fact, they include variables that are not possible to measure in the model type, such as political instability. The authors need to go deeper into the variables' justification to understand those included in the models. At the same time, it is necessary to differentiate between macro and micro factors (you are working with individual variables).

4) Hypothesis. The authors do not declare any hypothesis. Maybe that is a result of a weak literature review. To improve the text, the authors should elaborate hypotheses founded on literature. That action also will enhance the justification of the model.

5) SES variable. Why did the authors use this method to build SES and not another proxy for education level (usually correlated)? Authors include both in their model and household's head education level seems to perform better.

Stratification variables have an extensive discussion in social sciences. Even in epidemiology, some studies use this kind of indicator based on house assets (authors cited references); social sciences question the quality of these variables. The problem is visible regarding the results of this study because this variable behaves strangely in its link with food insecurity (see quintiles 2 and 4).

Otherwise, in the abstract, the authors mention that they work with mid and lower SES, but in their data, they built a wealth index, including high "wealthiness." How is this possible? That is not explained anywhere.

Two solutions are possible. Use education level (if the authors do not have other indicators). Better explain the index regarding the sample and its limitations.

Other comments:

Perhaps the word quintile is not the best word to describe grouping SES because it leads erroneously to think of a continuous variable with income.

Perception of food consumption does not seem justified insofar as it may have a high correlation with the food security scale, generating possible problems in the model.

I think the modeling is correct to have a sensitivity to seasonal aspects. Also, the work with longitudinal models.

It is not clear the inclusion of SES media in table 1.

Table 4 four does not contain all the reference categories listed, although it is possible to assume it.

6. PLOS authors have the option to publish the peer review history of their article (what does this mean?). If published, this will include your full peer review and any attached files.

Reviewer #1: No

Reviewer #2: **Yes: **Khushbu Mishra

Reviewer #3: No

---

## [Author Response · Author response to Decision Letter 0]

15 Oct 2022

JOURNAL REQUIREMENTS 

Response: We have reviewed the requirements and apply them to our manuscript.

Response: We will provide repository information for your data at acceptance

Response: We have included the full statement as indicated.

“KEP and MMTR. This work was supported by the National Institutes of Environmental Health Sciences at the National Institutes of Health (P01ES022844, R24ES028502), the US Environmental Protection Agency (RD83543601). The funders did not play any role in the conception or writing of this paper (or any others from the ELEMENT Project).”

Response: We have includede the funder´s role statement in the manuscript and cover letter.

REVIEWER´S COMMENTS

Reviewer #1: I have published a number of studies regarding the measurement of food insecurity, and I feel like the authors did an adequate, even good, job. There was a clear objective, the data allow them to achieve their objective, and they used scientifically valid statistical methods. I appreciate the authors keeping the manuscript succinct. For example, they could have spent pages and pages discussing the ELEMENT cohort but they prudently remark instead that it is described elsewhere and they provide that citation.

A couple of remarks: 

1) Use of the term “surge” in the title doesn’t bother me but most reviewers do not like that kind of language. 

Response: Our understanding is that the word “surge” is common in scientific literature. A quick search in PubMed yielded 19,623 results that use “surge” either in the title or abstract, and of those 2,869 use it in the title. If the reviewer does not mind, we would like to keep it as we think it is a good descriptor of the situation of food insecurity in the population “to rise suddenly to an excessive or abnormal value” (https://www.merriam-webster.com/dictionary/surge).

2) On line 126, what is a “secular” fluctuation in food security? May be a translation issue here. 

Response: We thank the reviewer for his/her comment. By secular we meant “long-term”, the word is common in econometric literature, especially in time-series analyses. However, we agreed to change that term to “periodic”, which we think is easier to understand. Line 154 “To evaluate periodic fluctuations in food insecurity during the study period…

3) On lines 136 – 137 you remark that you only retain covariates that show statistical significance. In my field that is highly discouraged and is referred to as a form of “p hacking”. People still do it all the time, you are just one of the few authors who admit it! If I were you though I would instead put your original estimations that also include the insignificant coefficients, lest someone use your paper as an example of what not to do. This isn’t a huge deal for me but it is for some in my field. 

Response: We thank the reviewer for this suggestion. We have removed said phrase from the statistical analysis sub-section and added the non-significant coefficients to table 4.

4) On Table 1 I don’t understand what the COV% column means. 

Response: The COV% column describes the characteristics of the sample during the pandemic in visit COV. We have added the word "visit COV" in the description of the table.

Reviewer #2: This study’s goal is to estimate the impact of the COVID-19 pandemic on the occurrence of food insecurity in a cohort of women living in Mexico City, and the socioeconomic characteristics associated with food insecurity severity. I can definitely see this paper making contribution to still ongoing issue that countries face due to COVID so it is a very relevant topic. However, the manuscript could be stronger with a few suggestions listed below:

Abstract

1) Typo in the abstract, Results portion

Response: We have corrected the typo in the Results portion of the abstract.

Study population and sample. 

2) The timings of the visits are different across the years. Is that a concern? How can that impact results depending on what season of the year/crop cycle it is in?

Response: We thank the reviewer for his/her comments. We do agree that the time of year can have an effect on food insecurity due to potential yearly cyclical changes in the economy. Therefore, we deemed very important to control the model for the month of the year to control for this potential confounding effect.

3) This should be written differently. “A total of 685 women had data on food security at least one data time point, while 432 women had information on all three time points.”

Response: We have modified the writing to improve clarity: line 118-119 “432 women were interviewed on all three time points, and 685 women were interviewed at one time point at least.”

Results

4) Whereas all other types of food insecurity seem to have increased, it is puzzling that Mild food insecurity decreased. This is also mentioned the in abstract. More explanation and intuition here could be helpful. 

Response: Overall food insecurity increased 12.2 and 7.6 percentage points at COV stage (53.8%) compared to previous stages: T1 (46.2%) and T2 (41.6%), respectively, which is a noteworthy result. But we intended to highlight the fact that the increase was concentrated mostly in the highest levels of food insecurity, as combined moderate-severe food insecurity increased 15.2 points from T2 (1.6%) to COV (16.8%) and12.8 points from T1 (4%) to COV stage (16.8%). Therefore, the decrease only in mild food insecurity is not a positive transition, since these results indicate that the sample population transited to the more serious levels of food insecurity in the COV stage. We agree that the reported result in the abstract can be unclear, therefore we modify the text for clarity: “Food insecurity (any level) increased from 41.6% duting the pre-pandemic period to 53.8% in the pandemic stage. This increase was higher in the combined severe-moderate food insecurity levels: from 1.6% pre-pandemic to 16.8% during the pandemic”. Additionally, we included the percentages of “Food insecurity (any level)” in Table 2 so all the results are shown and can be easily interpreted.

5) Line 131: missing comma

Response: We have inserted the comma.

6) the Table 1: since determinant seems to be pre and post pandemic timing, it would be better to present table 1 by the survey rounds, rather than quintiles (similar to table 2) 

Response: We thank the reviewer for this comment. We have two main objectives for our analysis: i) assess changes in food security using pre and post pandemic information of the same sample of families, and ii) assess if these changes affect more economically disadvantaged, that is, if families with lower socioeconomic status were more exposed to experience food insecurity during the sanitary crisis. Our hypothesis behind this objective is that demographic and economic characteristics of more vulnerable families increase the likelihood of food insecurity. Table 1 has the intention to show the reader which characteristics are associated with more disadvantaged families who are more likely to experienced food insecurity. On the other hand, most of the variables shown in Table 1 are unlikely to have changed after the pandemic, showing pre and post pandemic characteristics will not offer a lot of variation in the case.

7) Line 177: missing “the”

Response: We have inserted the missing article.

8) Figure 1 does not appear where its placed. 

Response: We have inserted Figure 1 in the document. 

Econometric method

9) Usually there are strong reasons behind why authors would choose to use FE versus RE models. In this paper, there is no justification or mention of why both are done? Also, there are no robustness checks of the models. 

Response: We agree with the reviewer that the choice of statistical model is utterly important. Unfortunately there is no “Hausman-like” test for Ordered Logit Regression implemented in the statistical package to help us objectively decide for the fixed or random effects ordinal logistic regression models. We believe that our choice to present both is actually a kind of robustness check, with different types of models yielding very similar results. Although fixed-effects models are theoretically known to be consistent, and thus likely preferable, the advantage of also presenting random-effects models is that they allow for time-fixed regressors to be included in the model. We have included this rationale in the revised statistical analysis section. (line 167-170) 

10) Since the month of data collection is different, I thought I saw somewhere that authors control for that, however, I don’t see that in Table 4. And how does controlling for months correlate with the survey rounds themselves? What is the correlation? 

Response: We apologize for our omission. The models did include months of the year as covariates, we now display the full set of coefficients in the revised Table 4. Months of the year were indeed significantly concordant with the timing of the surveys. The Spearman correlation of the month of the year with the timing of the survey rounds was 0.12 with a highly significant p-value (<0.001).

11) The authors state that they used a host of characteristics but they only present the ones that are statistically significant. Are the rest of the variables not presented because authors are running out of space in the paper? Mention of these variables in table notes would at least be expected so readers can see it at a glance. 

Response: We apologize for this omission. Please see response to reviewer 1 and our revised table, which now explicitly includes all tested regressors.

Discussion

12) “To our knowledge, this is the first study with panel data that estimated the magnitude of the COVID-19 impact on food insecurity in Mexico”. I might be wrong but I thought that the World Bank did it for many countries across the world and that should have included Mexico. 

Response: The studies from the WB did include Mexico; however, these studies do not rely on longitudinal information that uses changes in food security using the same set of families as in our case. To our knowledge, our study is the only example from Mexico with panel information. 

13) Line 217: grammar LB

Response: We have modified the writing to improve clarity: “Additional information on food insecurity in Mexico is available in the ENCOVID-19 survey (COVID-19 effects on the well-being of Mexican households survey), a nationally representative cross-sectional survey conducted monthly by telephone”.

Line 267: grammar LB

14) Response: The indicated text changed in response to other comments. We hope the new text has also improved in clarity. 

15) Line 269: the definition should be written early on since the study is about food insecurity

Response: We have placed the definition in the introduction.

Reviewer #3: The manuscript addresses a relevant issue for which evidence already exists in Mexico and other countries. Still, it contributes from a methodological perspective by using data from a panel study and incorporating a longitudinal analysis. Nevertheless, the article presents several problems regarding the representativeness of its results, context, dialogue with specialized international literature, hypothesis, and the SES variable.

16) Representativeness of results. The authors work with a cohort of women recruited from maternity hospitals in Mexico City (following the cited paper because it is not mentioned in this manuscript). How this cohort represents information about the general population? Why study this group could be interesting for understanding food insecurity? What are the limits of this study concerning this sample? The authors should explain more details about the recruitment, deep sample characteristics, and their limitations. 

 Response: We appreciate the reviewer’s question. First, we wholly agree with the reviewer in that the study sample is not representative of the Mexican or even Mexico City General Population, and this is acknowledged in the discussion section or the article. Although this could be seen a limitation, it is a limitation that is shared by most cohort studies. However, it should be noted that renowned epidemiologists have argued that representativeness is not only sometimes unnecessary, but it can even become a hurdle. (1) Others have pointed out that representativeness is necessary for obtaining good estimates of prevalence, but unneeded for analytical studies, in the sense that common objections to non-representative cohorts rarely cause bias beyond that which could be expected in representative samples. (2)

On the other hand, longitudinal (panel) data that originates from cohort studies has many advantages over the, often representative, data that is common in cross-sectional samples, of which we have cited several examples. It has been established that panel data, by blending the inter-individual differences and intra-individual dynamics can provide: 1. More accurate inference of model parameters; 2. Greater capacity for capturing the complexity of human behavior than a single cross-section or time series data. These include: 2.a Constructing and testing more complicated hypotheses; 2.b Controlling the impact of omitted variables; 2.c Uncovering dynamic relationships. 2.d Generating more accurate predictions for individual outcomes by pooling the data rather than generating predictions of individual outcomes using the data on the individual in question; amongst others. (3)In summary, we believe the strengths of the longitudinal data in terms of facilitating the finding of causal relationships, more than compensates for limitations in terms of representativeness of the sample.

1. Kenneth J Rothman, John EJ Gallacher, Elizabeth E Hatch, Why representativeness should be avoided, International Journal of Epidemiology, Volume 42, Issue 4, August 2013, Pages 1012–1014, https://doi.org/10.1093/ije/dys223

2. Lorenzo Richiardi, Costanza Pizzi, Neil Pearce, Commentary: Representativeness is usually not necessary and often should be avoided, International Journal of Epidemiology, Volume 42, Issue 4, August 2013, Pages 1018–1022, https://doi.org/10.1093/ije/dyt103

3. Hsiao, Cheng, Panel Data Analysis - Advantages and Challenges (May 10, 2006). IEPR Working Paper No. 06.49, Available at SSRN: https://ssrn.com/abstract=902657 or http://dx.doi.org/10.2139/ssrn.902657

17) The limits of this research imply that the authors must moderate some statements. For example, the abstract mentions that food insecurity soared in Mexico City in 2020 (in general), but in the same abstract, they mention that they work with data from mid and lower mid to low SES in Mexico City. If their sample came from this group of women, they must be cautious with generalizations. LB/LT

Response: We agree with this comment. We have modified the statem

---

## [Decision Letter · Decision Letter 1]

8 Dec 2022

PONE-D-22-12889R1A surge in food insecurity during the COVID-19 pandemic in a cohort in Mexico CityPLOS ONE

Dear Dr.  Bautista-Arredondo,

Thank you for submitting your manuscript to PLOS ONE. After careful consideration, we feel that it has merit but does not fully meet PLOS ONE’s publication criteria as it currently stands. Therefore, we invite you to submit a revised version of the manuscript that addresses the points raised during the review process.

Consider carefully the comments raised by the reviewer and the academic editor and make revise the manuscript accordingly. Also let a language editor review the work before submission. 

We look forward to receiving your revised manuscript.

Kind regards,

Sandra Boatemaa Kushitor, Ph.D.

Academic Editor

PLOS ONE

Journal Requirements:

Reviewers' comments:

Reviewer's Responses to Questions

**Comments to the Author**

1. If the authors have adequately addressed your comments raised in a previous round of review and you feel that this manuscript is now acceptable for publication, you may indicate that here to bypass the “Comments to the Author” section, enter your conflict of interest statement in the “Confidential to Editor” section, and submit your "Accept" recommendation.

Reviewer #1: All comments have been addressed

2. Is the manuscript technically sound, and do the data support the conclusions?

Reviewer #1: Yes

3. Has the statistical analysis been performed appropriately and rigorously? 

Reviewer #1: Yes

4. Have the authors made all data underlying the findings in their manuscript fully available?

Reviewer #1: Yes

5. Is the manuscript presented in an intelligible fashion and written in standard English?

Reviewer #1: Yes

6. Review Comments to the Author

Reviewer #1: (No Response)

7. PLOS authors have the option to publish the peer review history of their article (what does this mean?). If published, this will include your full peer review and any attached files.

Reviewer #1: **Yes: **Franklin Bailey Norwood

---

## [Author Response · Author response to Decision Letter 1]

13 Feb 2023

Editor comment: Please review your reference list to ensure that it is complete and correct. If you have cited papers that have been retracted, please include the rationale for doing so in the manuscript text, or remove these references and replace them with relevant current references. Any changes to the reference list should be mentioned in the rebuttal letter that accompanies your revised manuscript. If you need to cite a retracted article, indicate the article’s retracted status in the References list and also include a citation and full reference for the retraction notice.

Response: We reviewed our reference list. We confirm that is correct and complete. We have not included any retracted articles. We added one new reference to respond a reviewer´s comment: 36. Castillo MJ, Galicia M, Castellano F. Evolución del costo de los alimentos ante el COVID-19. Chile: Centro Latinoamericano para el Desarrollo Rural, 2021 15.

Reviewer #2

Comment: Replace with increased

Response: We replaced the term “soared” with “increased”, as suggested.

Comment: The appropraite recommendation based on the study is to suggest that the government or ministry of agric provide some cushioning against food insecurity. This study did not focus on the effects of food security

Response: Food security includes four dimensions: 1) food availability, 2) food access, 3) food utilization, and 4) stability. Alterations in any of the food security dimensions can result in a reduction of the quality and quantity of food consumed by household individuals. We would agree with the reviewer´s comment, if the insecurity problem observed in our sample population was due solely to food supply (food availability dimension), but this is not the case. By directly intervening in households, a program could help families with insecurity caused by disruption in all dimensions of insecurity: food availability, food access, food utilization, and stability. Based on this rationale, we propose to keep our original recommendation. 

Comment: Break into two sentences at least. This is too long

Response: We edited the text to separate into two sentences, as suggested. 

Comment: This statement is not supported. Rephrase. Look at these papers

Response: We rephrase the text as suggested.

Comment: if there was a question about employment status during the covid lockdown it Will be Good to include it

Response: Employment status was collected during the Covid stage with the question “whether the woman respondent had a job over the previous week”, and results are included in Table 1. Nevertheless, the variable "having a job last week" was not included in the models as it is most likely an intermediate variable between the "Pandemic" variable and food insecurity. In other words, at least part of the effect of the pandemic on food insecurity is likely to be due to unemployment, therefore adjusting for said variable would cause an underestimation of the true effect. This information has been added to the updated "statistical analysis" sub-section.

Comment: you did not present any framework

Response: By conceptual framework we are referring to the background knowledge of food insecurity drivers that was included in the “Introduction”. To clarify we edited the text accordingly. 

Comment: provide an explanation of why this was done

Response: The reason for inclusion of the time variable (months of the year is described in the previous paragraph (lines 154-158). The rest of the covariates were deemed necessary to include as previous knowledge on food insecurity as per our literature review has uncovered them as clear drivers of the phenomenon. Current employment was not included in any model as it was deemed an intermediate variable. Models eventually excluded the marital status covariate as it presented low variation (less than 10% of the households had a single mother) and was not significantly related to the outcome. This explanation regarding employment was added to the updated "statistical analysis" sub-section.

Comment: specify the proportion who were married and those who were cohabiting. These are important indicators

Response: We separated the proportion of “married” and “cohabiting” women and the proportions are now in the text, Tables 1 and 3.

Comment: what does this mean

Response: “COV” indicate the Covid visit data point. To avoid confusion we deleted the term from the Table since the data included is already specified in the title of the Table. 

Comment: these variables are missing in the models; provide an explanation

Response: The variable "having a job last week" was not included in the models as it is most likely an intermediate variable between the "Pandemic" variable and food insecurity. In other words, at least part of the effect of the pandemic is likely to be due to unemployment, therefore adjusting for said variable would cause an underestimation of the true effect. Marital status was initially evaluated but ultimately excluded of the models due to lack of statistical significance, likely due to its low variation. This information has been added to the updated "statistical analysis" sub-section.

Comment: Include the total wealth quintiles

Response: The wealth index was categorized into SES quintiles, this is explained in lines 131-148. The value of the index has no intrinsic value, it only indicates if a household is richer than other. That is, the value of the index for each household only ranks households (we know if household “X” is richer than household “Y”, but we do not know by how much household “X” is richer than household “Y”). To know “by how much” we would need to use an absolute measure of economic wellbeing such as income or expenditures, but this type of information is not available in our data.

Below we are including the mean value of each SES quintile. 

Variable n Mean(SD)

 SES (continuous) 446 0.037(1.51)

 SES (quintiles) 

1 89 -2.47(1.28)

2 88 -0.22(0.24)

3 88 0.37(0.13)

4 89 0.87(0.16)

5 92 1.58(0.32)

Comment: be consistent with the unit of analysis; use either women or households throughout the study

Response: OK, we replace the unit of analyses consistently with “households”.

Comment: where should Table 2

Response: We moved Table 2 to place it correctly.

Comment: this should be T3. In the methods you mentioned that the data was collected between 2020 and 2021. You only have 2020 here 

Response: We added the year “2021” to the heading of Table 2.

Comment: place the tables at the appropriate location

Response: We have placed all Tables in the appropriate location. 

Comment: kindly use Mar, Sept, Mar etc instead of mar, sep

Response: We modified the Figure labels as suggested.

Comment: did you check whether households that were food insecure in T1 and T2 were the same households that were food insure during COV 

Response: Yes, we checked the transition or change of food insecurity from T2 to the COV stage. Results showed no changes in 52.6% of households (34.5% remained in food security and 18.1% remained in mild food insecurity). 36.1% households changed for the worst (24.2% were food secured in T2 and for the COV stage showed some level of insecurity, and 11.9% changed from mild food insecurity to a more serious levels of insecurity). Finally, 11.3% improved their food insecurity at some level.

Comment: some of the p’s are caps others are not

Response: We checked all the p´s and made sure the lower case use is consistent. 

Comment: marital status is missing from this model

Response: As explained in the updated "statistical analysis" section, marital status was eventually excluded of all models as it was not significantly related to food insecurity, likely due to its low variation.

Comment: month data was collected

Response: Yes, Figure 1 shows the patterns by month in which the data was collected, which is why it is important to adjust for this variable.

Comment: delete based on the papers I have showed you

Response: We deleted the sentence as suggested

Comment: ref

Response: We added one new reference to respond a reviewer´s comment: 36. Castillo MJ, Galicia M, Castellano F. Evolución del costo de los alimentos ante el COVID-19. Chile: Centro Latinoamericano para el Desarrollo Rural, 2021 15.

Comment: considering that 2022 is ending, revise this

Response: We edited the text as suggested.

---

## [Decision Letter · Decision Letter 2]

30 Jun 2023

PONE-D-22-12889R2A surge in food insecurity during the COVID-19 pandemic in a cohort in Mexico CityPLOS ONE

Dear Dr. Tellez-Rojo,

Thank you for submitting your manuscript to PLOS ONE. After careful consideration, we feel that it has merit but does not fully meet PLOS ONE’s publication criteria as it currently stands. Therefore, we invite you to submit a revised version of the manuscript that addresses the points raised during the review process.

We look forward to receiving your revised manuscript.

Kind regards,

Octavio Gómez-Dantés, MD, MPH

Academic Editor

PLOS ONE

Journal Requirements:

Reviewers' comments:

Reviewer's Responses to Questions

**Comments to the Author**

1. If the authors have adequately addressed your comments raised in a previous round of review and you feel that this manuscript is now acceptable for publication, you may indicate that here to bypass the “Comments to the Author” section, enter your conflict of interest statement in the “Confidential to Editor” section, and submit your "Accept" recommendation.

Reviewer #1: All comments have been addressed

Reviewer #4: (No Response)

2. Is the manuscript technically sound, and do the data support the conclusions?

Reviewer #1: Yes

Reviewer #4: No

3. Has the statistical analysis been performed appropriately and rigorously? 

Reviewer #1: Yes

Reviewer #4: I Don't Know

4. Have the authors made all data underlying the findings in their manuscript fully available?

Reviewer #1: Yes

Reviewer #4: No

5. Is the manuscript presented in an intelligible fashion and written in standard English?

Reviewer #1: Yes

Reviewer #4: No

6. Review Comments to the Author

Reviewer #1: (No Response)

Reviewer #4: 1. The intro is too long and it is not clear what is the issue under examination. It would be useful to see better structured narrative that is going from the general on food insecurity to the specific issue addressed in this manuscript. Unless you are addressing in a differentiate way the dimensions of food insecurity, there is no need to describe them.

2. How you are measuring impact? Your are describen a before and after analysis, that is not an estimation of the impact.

3. It is not clear what is the valued added of your study given that there are already data available on the topic in Mexico.

4. Additional details on the cohort is required in this paper.

5. Data was collected using different procedures. The period for data collection was not the same for the three data points; also, data was collected in different months of the year. How that could affect the answers? If each of the rounds was collected in a different month and different time frame, it is not clear how your models could correct for that. Please expand the details of your methods to clarify this.

6. The expected value refers to yes/no food insecurity? If that is the case, how it can be above 1? Why the CI of the COVID period is wider? Which the CI for all periods is wider at the beginning and at the end?

7. As your regression is using all observations, it not clear how it can report “protective factors” during the pandemic? Did you interacted those factors with the period?

8. Your discussion should reflect your results only; in your current version, while it acknowledges the lack of generalization, then generalizes.

7. PLOS authors have the option to publish the peer review history of their article (what does this mean?). If published, this will include your full peer review and any attached files.

Reviewer #1: **Yes: **Franklin Bailey Norwood

Reviewer #4: No

---

## [Author Response · Author response to Decision Letter 2]

8 Sep 2023

Reviewer #4: 1. The intro is too long and it is not clear what is the issue under examination. It would be useful to see better structured narrative that is going from the general on food insecurity to the specific issue addressed in this manuscript. Unless you are addressing in a differentiate way the dimensions of food insecurity, there is no need to describe them.

Response: We have edited the introduction according the reviewer´s comments. We deleted not essential information and the description of the four dimension of food insecurity that a previous reviewer asked to be added. The current version content is as follows. The first paragraph describes the effects of COVID on labor and income, which are major determinants of food insecurity. The second defines food insecurity, and explains how COVID affects food insecurity. The third presents estimates of food insecurity at the beginning of the pandemic as reference. The fourth states the objective of the article. 

2. How you are measuring impact? Your are describen a before and after analysis, that is not an estimation of the impact.

Response: We understand the reviewer’s concern regarding causal language. We have reviewed the use throughout the paper and consider the use of the term “impact” as appropriate. The aim of the study is to quantify the causal effect of the COVID-19 pandemic on food insecurity in the cohort of study. We cannot do a randomized trial to assess this. Our observational data is our only opportunity to estimate this effect. Because of the lack of randomization our results are associations that may not reflect causation, but the aim of our observational study is indeed causal, and we believe the language we use is acceptable. (see Hernán MA. The C-Word: Scientific Euphemisms Do Not Improve Causal Inference From Observational Data. Am J Public Health. 2018) 

3. It is not clear what is the valued added of your study given that there are already data available on the topic in Mexico. 

Response: The main added value of our article is the longitudinal panel data collected at two time points before the pandemic, and one-time point during the COVID pandemic in the same households. This strength is mentioned in the article, describing how most of the literature on the topic comes from cross-sectional analyses. Longitudinal (panel) data that originates from cohort studies, as our article, has many advantages over the, often representative, data that is common in cross-sectional samples. It has been established that panel data, by blending the inter-individual differences and intra-individual dynamics, can provide:

• More accurate inference of model parameters;

• Greater capacity for capturing the complexity of human behavior than a single cross-section or time series data. These include:

o Constructing and testing more complicated hypotheses

o Controlling the impact of omitted variables

o Uncovering dynamic relationships;

o Generating more accurate predictions for individual outcomes by pooling the data rather than generating predictions of individual outcomes using the data on the individual in question; amongst others.

In summary, we believe the strengths of the longitudinal data in terms of facilitating the finding of causal relationships more than compensates for limitations in terms of representativeness of the sample. 

Additionally, we'd like to kindly remind that PLOS ONE's publication criteria focus on the technical and scientific rigor of the work rather than its perceived novelty or broader significance. Our manuscript adheres to these criteria by presenting a methodologically sound study. We believe that every contribution that meets these rigorous scientific standards has value in the scholarly ecosystem, and we are confident that our work meets PLOS ONE's requirements in this regard.

4. Additional details on the cohort is required in this paper.

Response: We added additional details on the cohort. 

5. Data was collected using different procedures. The period for data collection was not the same for the three data points; also, data was collected in different months of the year. How that could affect the answers? If each of the rounds was collected in a different month and different time frame, it is not clear how your models could correct for that. Please expand the details of your methods to clarify this.

Response: We thank the reviewer for his/her comments. We do agree that the time of year can have an effect on food insecurity due to potential yearly cyclical changes in the economy. Therefore, we deemed very important to include month of the year as a covariate in the models, to control for this potential confounding effect.

6.1 The expected value refers to yes/no food insecurity? If that is the case, how it can be above 1? 

Response: We thank the reviewer for raising this important point. The expected value can indeed be above 1 because we are not modeling a binary 'yes/no' outcome for food insecurity. Instead, we are using a food insecurity scale that ranges from 0 to 3. To avoid any confusion, we have clarified this in the statistical analysis section of the revised manuscript. 

6.2 Why the CI of the COVID period is wider? 

Response: We thank the reviewer for her/his insightful question regarding the wider confidence interval (CI) observed during the COVID period. Based on our data, the variability in the food insecurity scale increased during this time, as evidenced by the following:

o Increased Levels of Moderate and Severe Food Insecurity: We observed a significant increase in moderate (from 1.0% in T2 to 9.0% in the COVID period) and severe food insecurity (from 0.6% in T2 to 7.8% in the COVID period).

o Decrease in No Insecurity: The percentage of households with no food insecurity dropped from 58.4% in T2 to 46.2% during the COVID period.

o Mild Food Insecurity: Although mild food insecurity decreased slightly, the pronounced changes in the other categories contribute to the overall increased variability.

This increased variability most likely contributed to the wider CI during the COVID period, as greater variability generally leads to less precise estimates.

6.3 Which the CI for all periods is wider at the beginning and at the end?

Response: The wider confidence intervals observed at the beginning and end of each study period are consistent with established statistical principles that are also applicable to local polynomial smoothing methods used in this study. Data points at the extremes have greater 'leverage' and can thus disproportionately influence the model's parameters, leading to increased uncertainty. Additionally, the density of data points is often lower at the extremes, which can compromise the model's ability to make precise estimates. These factors collectively contribute to the wider confidence intervals observed. For a deeper understanding, the reader is referred to 'Applied Linear Statistical Models' by Kutner et al. (2005), which discusses the influence of leverage points and data density on regression estimates, including local polynomial methods.

7. As your regression is using all observations, it not clear how it can report “protective factors” during the pandemic? Did you interacted those factors with the period?

Response: The query about reporting 'protective factors' during the pandemic is insightful. In this study, the protective odds ratios are derived from a multiple ordinal regression model that adjusts for various predictors, including study period and socioeconomic status, rather than from interaction terms with the study period. The analysis suggests that these factors are protective against food insecurity in a more general sense, and their protective effect is not specifically tied to the pandemic period. Therefore, it can be reasoned that these factors serve as protective elements against food insecurity regardless of the time frame. Clarifications will be made in the wording of the results section to better articulate this point, specifically, we removed the “during the pandemic” phrase from lines 212-213 in page 13 of the revised version, as that was indeed imprecise.

8. Your discussion should reflect your results only; in your current version, while it acknowledges the lack of generalization, then generalizes.

Response: The reviewer's point about the Discussion section is well-received. While it is our view that acknowledging a study's limitations is not mutually exclusive with discussing its broader implications, we appreciate the concern regarding the perception of generalization. To directly address this, the manuscript’s concluding remarks were refined to better articulate the scope of the study's implications. Specifically, the concluding sentence was updated to state: “Our findings contribute to the growing body of evidence suggesting that well-designed and focalized programs are especially crucial during pandemics or natural disasters, to prevent families from the adverse health and well-being effects associated with food insecurity.” This adjustment aims to clearly situate our recommendations within the broader context and specify conditions where they are most relevant, thereby addressing concerns about generalization.

---

## [Decision Letter · Decision Letter 3]

11 Jan 2024

A surge in food insecurity during the COVID-19 pandemic in a cohort in Mexico City

PONE-D-22-12889R3

Dear Dr. Tellez-Rojo,

We’re pleased to inform you that your manuscript has been judged scientifically suitable for publication and will be formally accepted for publication once it meets all outstanding technical requirements.

Kind regards,

James Mockridge

Staff Editor

PLOS ONE

Reviewers' comments:

Reviewer's Responses to Questions

**Comments to the Author**

1. If the authors have adequately addressed your comments raised in a previous round of review and you feel that this manuscript is now acceptable for publication, you may indicate that here to bypass the “Comments to the Author” section, enter your conflict of interest statement in the “Confidential to Editor” section, and submit your "Accept" recommendation.

Reviewer #2: All comments have been addressed

Reviewer #3: (No Response)

Reviewer #4: All comments have been addressed

2. Is the manuscript technically sound, and do the data support the conclusions?

Reviewer #2: Yes

Reviewer #3: Yes

Reviewer #4: Yes

3. Has the statistical analysis been performed appropriately and rigorously? 

Reviewer #2: Yes

Reviewer #3: Yes

Reviewer #4: Yes

4. Have the authors made all data underlying the findings in their manuscript fully available?

Reviewer #2: Yes

Reviewer #3: No

Reviewer #4: Yes

5. Is the manuscript presented in an intelligible fashion and written in standard English?

Reviewer #2: Yes

Reviewer #3: Yes

Reviewer #4: Yes

6. Review Comments to the Author

Reviewer #2: The paper has come a long way from its first submission in a positive manner. I wish the authors all the best!

Reviewer #3: The manuscript addresses a relevant issue, but evidence already exists in Mexico and other countries. The text only contributes from a methodological perspective, using data from a panel study and incorporating a longitudinal analysis. However, as I noted before, there are no substantial novelty and contribution.

The manuscript continues with a relevant lack in the literature review.

In addition, I still need to find in the text significant changes regarding my original comments.

Reviewer #4: (No Response)

7. PLOS authors have the option to publish the peer review history of their article (what does this mean?). If published, this will include your full peer review and any attached files.

Reviewer #2: No

Reviewer #3: No

Reviewer #4: No

---

## [Editor Report · Acceptance letter]

29 Apr 2024

PONE-D-22-12889R3 

PLOS ONE

Dear Dr. Tellez-Rojo, 

I'm pleased to inform you that your manuscript has been deemed suitable for publication in PLOS ONE. Congratulations! Your manuscript is now being handed over to our production team.

Kind regards, 

on behalf of

Dr James Mockridge 

Staff Editor

PLOS ONE